# Accuracy of Buffy Coat in the Diagnosis of Disseminated Histoplasmosis in AIDS-Patients in an Endemic Area of Brazil

**DOI:** 10.3390/jof5020047

**Published:** 2019-06-09

**Authors:** Terezinha M. J. Silva Leitão, Antonio M. P. Oliveira Filho, José Evaldo P. Sousa Filho, Bruno M. Tavares, Jacó R. L. Mesquita, Luís Arthur B. G. Farias, Rosa S. Mota, Mathieu Nacher, Lisandra S. Damasceno

**Affiliations:** 1Departamento de Saúde Comunitária, Faculdade de Medicina, Universidade Federal do Ceará, Rodolfo Téofilo, Fortaleza, Ceará 60430-160, Brasil; antonio_mponte@hotmail.com (A.M.P.O.F.); j.evaldo.filho@gmail.com (J.E.P.S.F.); luisarthurbrasilk@hotmail.com (L.A.B.G.F.); rosamarias688@gmail.com (R.S.M.); 2Hospital São José de Doenças Infecciosas, Parquelândia, Fortaleza, Ceará 60455-610, Brasil; brunomelotavares@gmail.com (B.M.T.); jacomesquita@hotmail.com (J.R.L.M.); lisandraserra@yahoo.com.br (L.S.D.); 3Centre d’Investigation Clinique 1424 Antilles-Guyane, Inserm, Centre Hospitalier de Cayenne, 97 300 Cayenne, France; mathieu.nacher66@gmail.com; 4Faculdade de Medicina, Centro Universitário Unichristus, Cocó, Fortaleza, Ceará 60190-060, Brasil

**Keywords:** buffy coat, diagnosis, histoplasmosis, AIDS

## Abstract

The buffy coat is obtained routinely for disseminated histoplamosis (DH) diagnosis in Ceará, Brazil. The aim of this study is to describe the accuracy of staining smears for *Histoplasma* in the buffy coat of AIDS-patients with DH. From 2012–2013, all results of stained buffy coat smears and culture for fungi performed at São José Hospital were recorded. In total, 489 buffy coats of 361 patients were studied; 19/361 (5.3%; 95%CI = 2.9–7.6%) had positive direct examination stained smears for *Histoplasma* and 61/361 (16.9%; 95%CI = 13.0–20.8%) had growth in culture. For those with positive *Histoplasma* cultures, the CD4 count was significantly lower (139.3 vs. 191.7cells/µL; *p* = 0.014) than others, and death was 18%. The sensitivity and specificity of stained smears was 25.9% and 100%, respectively. A second test, performed up to 36 days from the first one, increased the sensitivity of stained smears to 32.2%. Stained smears of buffy coat have low accuracy; nonetheless, they are easy to perform and can give a quick diagnosis in low-resource endemic areas. Despite the decrease in mortality, it is not yet to the low levels observed in areas that have better and more efficient methods.

## 1. Introduction

Histoplasmosis is a widely distributed disease caused by the dimorphic fungus *Histoplasma capsulatum* [1]. It is considered the most common endemic mycosis in the United States and in certain areas of Mexico and Central and South America [1,2]. Two varieties of *H. capsulatum* are known: var *capsulatum* and var *duboisii*. Both are indistinguishable in their mycelian form, but differ in their yeast; in var *duboisii* (restrict to the tropical areas of Africa), the cells have thicker walls and are larger, consequently, not seen inside leukocytes as the var *capsulatum* [3].

In the HIV era, disseminated histoplasmosis (DH) has emerged as a public health problem in endemic areas, notably those with low resources, mainly affecting AIDS-patients with severe immunosuppression. In Ceará state, located in northeastern Brazil, DH is an important cause of hospitalization and death in HIV-infected patients [4]. Damasceno et. al. published a study evaluating 145 AIDS-patients with DH and showing an overall fatality rate of 30.2% [5].

In many parts of the world, DH diagnosis is still a challenge for many physicians, due to the lack of sensitive/specific tests, the low degree of suspicion, and unavailability of the best treatment. Thus, it remains a neglected disease [6,7]. The diagnosis of DH can be made through staining smears, culture, serology, histopathology, antigen detection, or molecular analysis. Currently, the test with the greatest combination of rapidity, low-cost specificity, and sensitivity in disseminated cases is the identification of *Histoplasma* antigens in blood or urine samples [8,9]. However, antigen detection is not broadly available; consequently, the diagnostic approaches rely on slow methods, such as fungal culture or those with low sensitivity such as staining smears, and presumptive diagnosis remains necessary for case management in low- or medium-resource countries [10,11].

The microscopic analysis of the buffy coat was initially used at the São José Hospital for the diagnosis of Kala-azar in AIDS-patients in the early 90s. However, instead of the *Leishmania*, the tested samples revealed an elevated frequency of structures compatible with *Histoplasma*, which were all confirmed by culture (unpublished data). This method was then routinely used for the diagnosis of the DH, and the State of Ceará began to be recognized as an area of high endemicity for histoplasmosis. 

The aim of this study is to describe the experience of the São José Hospital using buffy coat as a diagnostic tool for DH in HIV-patients, and to determine the sensitivity and specificity of this test in that specific population.

## 2. Materials and Methods

### 2.1. Study Population and Research Design

This was a cross-sectional study conducted at Hospital São José, a public infectious diseases reference hospital for the HIV-positive population of Ceará State, in Northeastern Brazil.

During the 2012 to 2013 period, all results of stained smears and culture of buffy coat for fungi performed at the Hospital São José laboratory were recorded. Only microscopically examined samples of HIV-positive patients with cultural results for fungi were included. Searching for *Histoplasma* in the buffy coat (direct examination of smear and culture) of HIV-patients with fever is a standard practice for the diagnosis of DH at this hospital.

Patient’s medical records were reviewed for data entry. The variables analyzed were age, sex, CD4 counts, death, cytology, and culture results for fungus. Sequential samples were considered within a maximum interval of 36 days and were analyzed considering the results in the first and second testing. Histoplasmosis was considered as the same episode when it occurred within a period of less than a year. Missing culture records, contaminated cultures, and growth of other fungal species led to the exclusion of the samples from the present study.

### 2.2. Buffy Coat Assay

Buffy coat contains most of the white blood cells and platelets. It can be obtained after collection of four milliliters of peripheral blood in ethylenediaminetetra-acetic acid (EDTA) tube. After centrifugation of the blood sample in macrohematocrit tube in 2000 rpm for 10 min, a thin layer in between the plasma and red blood cells (the buffy coat) was transferred to a sterile glass tube. One drop of the buffy coat was placed over two glass slides and stained by Giemsa for microscopic examination. Meanwhile, another two drops were inoculated into two tubes, one containing Sabouraud Dextrose Agar (SDA) and another SDA tube with cycloheximide for agent isolation, both maintained for up to 30 days at 30 °C temperature. Smears for *Histoplasma* were considered positive when yeasts of 2–5 micrometers were identified in/or surrounding the leucocytes, and the cultures were considered as positive, if mycelium-phase as hyphae and tuberculate macroconidia were seen after staining with cotton blue. *Histoplasma* positive cultures are routinely sent to the Centro de Especialidades em Micologia Médica, a local reference laboratory for fungi of the Federal University of Ceará that confirms the identification by micromorphology and the conversion from the mold to the yeast phase; if necessary, the molecular method is performed.

### 2.3. Statistical Analysis

The IBM SPSS base 20 was the statistical software for data analysis. The Mann Whitney test was used to compare numeric variables and the Fisher’s exact test for contingence tables 2 × 2. For the concordance analyses, the agreement coefficient of Kappa was applied. The sensitivity and specificity analyses were also obtained. The level of significance considered was 5%.

### 2.4. Ethical Aspects

This study was part of broader research on DH that was approved by the Ethical Review Board of São José Hospital of Infectious Diseases, Fortaleza, Ceará, Brazil. This study follows the rules of the Declaration of Helsinki of 1975, revised in 2013. 

## 3. Results

### 3.1. Study Design

During the study period (2012–2013), the São José Hospital laboratory performed 608 buffy coat analyses, 119 of which were excluded due to HIV-negative results (17), contamination (6)/ unavailable information (27) or growth of other fungi (3) in culture (all direct examinations of stained smears were negative), and missing data (66). In total, 489 buffy coat samples of 361 patients were studied. In 19 (3.9%; 95%IC: 1.9–5.9%) of the 489 samples, the direct examination of stained smears was positive for *Histoplasma*, and in 73 (15%; 95%IC: 11.3–18.7%) this fungus was isolated by culture. Of the 361 patients, 19 (5.3%; 95CI = 2.9–7.6%) had positive direct examination stained smears for *Histoplasma* and 61 (16.9%; 95%CI = 13.0–20.8%) had growth in culture.

### 3.2. Population Baseline

The mean age of the 361 patients was 35.7 years; males were the majority (76.5%), and the overall mortality was 20.2% (18% for the subgroup of patients with *Histoplasma* positive cultures). The CD4+ T lymphocyte count in the *Histoplasma* positive culture group was significantly lower than the negative culture group (139.3 vs. 191.7 cells/µL; *p* = 0.014). 

Data concerning the distribution by age, sex, death, and CD4 counts among the patients with positive or negative culture for *Histoplasma* are displayed in Table 1.

### 3.3. Sequential Tests

In 99 patients, more than one buffy coat examination was obtained. Of these, 78 had two samples tested, and in 21 more than three buffy coats were obtained (14, six, and one patients had three, four, and five exams collected, respectively). None of the stained smears became positive in the third or more collected samples; two cultures became positive only in the third exam.

Two of the 99 patients had more than one year between the tests and were not considered to be the same infectious episode. Of the 97 remaining patients, 64 tests were done within an interval of less than 36 days and were analyzed as sequential tests. 

The histoplasmosis prevalence in the sequential tests was 7.8% (95%IC: 3.4–17.0%) for the direct examination of stained smears and 21.9% (95%IC: 13.5–33.4%) for culture. The mean time interval between sequential buffy coat analysis was 10.8 days (SD: 9.4). In these sequential tests, three negative stained smears in the first test became positive in the second one (mean of 13.6 days; SD = 13); of the 14 positive cultures, two were positive only in the second sample (mean 8.2 days; SD = 9.8). Ten out of 12 cases had fungal growth in the first culture performed and remained positive in the second one, after a mean of 8.1 days (SD = 9.8). 

### 3.4. Accuracy Tests

All positive direct examinations of stained smears for *Histoplasma* were confirmed by culture. Compared to stained smears, the buffy coat culture increased the prevalence of histoplasmosis in 11.1% of those with one test.

Comparing with culture (the gold standard for histoplasmosis diagnosis, with a prevalence of 15% in one test, in this study), the sensitivity and specificity of stained smears was 25.9% and 100%, respectively. When the sequential test results were included in the analysis, the sensitivity of the stained smear increased to 32.2% (Table 2).

The Kappa analysis agreement between the stained smears and the culture performed in the first test was low, although significant (Kappa [K] = 0.37; 95%IC: 0.29–0.45).

## 4. Discussion

The high prevalence of *Histoplasma* detected in cultures of buffy coat in the present study confirms the high endemicity of this fungus in Ceará, in Northeast of Brazil, as described in previous studies [5,11,12]. The low sensitivity of stained smears for *Histoplasma* diagnosis has also been described before [10,13]; however, in settings where the most sensitive methods are not available [14] (the *Histoplasma* antigen test), which is the case in the majority of endemic areas around the world, mainly in poor countries, the direct examination of buffy coat stained smears for fungus could help decision-making by physicians, based on the high positive predictive value-PPV. Culture must always be requested due to its better sensitivity and for subsequent case confirmation.

*Histoplasma* has a tropism for mononuclear cells [15], and concentrating them in the buffy coat increases the probability to detect the agent. In the absence of better tests, the buffy coat is more easily obtained than the bone marrow and is not impeded by blood dyscrasia. All microscopically identified *Histoplasma* at Hospital São José laboratory were confirmed by culture. Despite the operator dependence of direct examination of stained smears examination, in this specific endemic area, the lab-personnel were confident in identifying *Histoplasma*, and misdiagnoses with *Candida*, *Taloromyces marneffei*, and *Leishmania* were not observed.

The diagnosis of histoplasmosis mainly relies on clinical presentation and test availability. The disseminated presentation when associated with skin or mucosal lesions can lead to a quick diagnosis by direct examination of the skin or mucosal smear; however, this takes place in an average of 30% [11] of the cases, and may be related to dissemination and severity (usually linked to late diagnosis) and consequently with death. 

In areas with limited resources, the approach of DH diagnosis is mostly presumptive [10,11]. Elevated lactate dehydrogenase, pancytopenia, hyperferritinemia, and high aspartate aminotransferase are nonspecific biologic signs that orient the decision-making regarding the treatment of suspected AIDS-patients [10,11]. In this type of setting, such as the Hospital São José, the buffy coat stained smears can be an important tool. In a study conducted at Hospital São José in 2006, 48 cases of DH were detected through an active search for AIDS-patients suspects of DH in hospital wards. Biological markers such as hepatomegaly, CD4 count ≤ 75 cells/µL, LDH level ≥ 5x the UNL, and maculopapular or papular lesions were significantly associated with DH. In that study, the diagnosis of DH was established from direct examination of stained smears (mainly buffy coats) and culture in 64.5% and 31.2% of cases, respectively. Another finding from this cited research was the reduced mortality rate (20.8%) when compared with the 42.3% previously reported at the same setting [12]. According to the authors, the decline was a consequence of the active search for DH-suspect patients, which improved the diagnosis and treatment, a finding that was confirmed in the present study, in that the death number of patients with positive direct examination of buffy coat stained smears and culture was 18%.

The recovery of the *Histoplasma* in culture of biologic specimens, and the histopathologic or direct microscopic demonstration of characteristic intracellular yeast forms in a phagocytes, in a peripheral blood smear, or in tissue macrophages, are considered criteria for the proven diagnosis of invasive histoplasmosis [16]. Otherwise, antigen detection provides a rapid, non-invasive, and highly sensitive method for diagnosis as well as a useful marker of treatment response [8]. Molecular methods with improved sensitivity on clinical specimens have began to be used, as published in reported cases [17].

Sequential analyzes of buffy coat showed little benefit. However, the samples used for the analysis were not collected at standardized time intervals, and it is possible that this could have interfered with the findings. Information regarding the use of antifungal therapy during the samples collection was not obtained.

In *Histoplasma*-endemic areas, a CD4 count < 150 cells/µL places AIDS-patient at high risk of DH and is considered an indication for prophylactic treatment [18]. In the current study, patients with DH had lower CD4 counts, indicating advanced immunosuppression. Nonetheless, differences in mortality were not observed between those with and without DH, although we cannot discard the possibility of histoplasmosis cases in the culture-negative group since the most sensitive method for case detection was not used. 

Histoplasmosis is a substantial contributor to mortality in people with HIV in Ceará. In several published articles about histoplasmosis in AIDS-patients from that state, the mortality rate in the last two decades was not less than 20% [5,11,12,19,20]. A systematic review of the frequency and mortality of histoplasmosis among people with HIV in Latin America was recently investigated. The authors found that histoplasmosis was frequent and mortality was high despite the use of HAART. Low CD4 counts, delayed HAART initiation, and poor adherence were related to increased incidence, poor prognosis, and increased mortality, respectively [21]. 

The lower death rate observed in the present study, compared to earlier years, is believed to be related to the increasing awareness of the local physicians to identify DH cases, as already reported for HIV-associated histoplasmosis in French Guiana [22]. The systematic request of buffy coat for direct examination of stained smears and culture in AIDS-patients suspects of DH is a usual practice in São José Hospital, and this may explain the frequent identification of this organism in that area and not in the nearby states that share the same ecosystem.

The concordance between the positive smears and culture of buffy coat of the present study was low. In contrast, testing a promising MiraVista lateral flow assay for *Histoplasma* detection showed high sensitivity (95%), good specificity (82%), and great agreement comparing human eye and automated reader [23]. Furthermore, as shown in a recent review, the diagnostic accuracy of *Histoplasma* antigen tests in the context of advanced HIV-disease demonstrated the good performance of different antigen detection tests [14].

In the present study, the culture of buffy coat increased fungus detection by 11%. It is a lengthy method and is not suited to rapid decision-making. At Hospital São José, culture is used mainly for diagnosis confirmation, after the commencement of a specific treatment. Based on the percentage (70.8%) of positive culture for *Histoplasma* in AIDS-patients with DH [24], the sensitivity finding of direct examination of buffy coat stained smears obtained in the present study was probably overestimated, and this is a limitation of the study, but one that can be overcome by future researches. 

Reducing diagnostic delays and therapy initiation is needed to further decrease mortality, mainly in patients with advanced HIV-disease (the leading risk factor for disseminated histoplasmosis). Equipping all the major hospitals in Latin America with simple, rapid diagnostic tests and antifungals to better diagnose and treat disseminated histoplasmosis in AIDS-patients by 2020 was a goal established in an international meeting in 2015, which joined several researchers from different countries of Latin America. The primary interest in obtaining rapid, affordable diagnostic tests is to improve detection and ultimately reduce the burden of histoplasmosis mortality in patients with advanced HIV-disease.

In summary, despite of the low accuracy of the stained smears of buffy coat, in the absence of more sensitive methods, this test is easily performed and, when positive, could help decision-making by assistant professionals, and probably decrease mortality, but not yet to the low levels observed in areas that have better and more efficient methods.

## 5. Conclusions

Stained smears of buffy coat have low accuracy; nonetheless, they are easy to perform, can give a quick diagnosis in low-resource endemic areas, and can help reducing mortality.

## Figures and Tables

**Table 1 jof-05-00047-t001:** Variables evaluated in the buffy coat culture for the diagnosis of disseminated histoplasmosis in AIDS-patient in an endemic area of South America. Fortaleza, Ceará, Brazil, 2012–2013.

	*Histoplasma* Detected by Culture		
Variables	Positive	Negative	*p*
Age mean (SD)	33.9(10.7)	37.4(11.5)	0.109
Sex			
Male (%)	49 (80.3%)	227 (75.7%)	0.510
Female (%)	12 (19.7%)	73 (24.3%)	
CD4 cells/µL, mean (SD)	139.3 (213.3)	191.7 (230.1)	0.014
Death	11 (18.0%)	62 (20.7%)	0.729

**Table 2 jof-05-00047-t002:** Accuracy of staining smears of buffy coat compared with culture in the diagnosis of disseminated histoplasmosis in AIDS-patient in an endemic area of South America. Fortaleza, Ceará, Brazil, 2012–2013.

Accuracy Test						
Buffy coat smear	Sensitivity	Specificity	PPV ^1^	NPV ^2^	FP ^3^	FN ^4^
One test	25.9%	100.0%	100.0%	88.5%	0.0%	11.5%
First and Second test	32.2%	100.0%	100.0%	88.3%	0.0%	11.7%

^1^ Positive predictive value; ^2^ negative predictive value; ^3^ false positive; and ^4^ false negative.

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
