# Peer review of "Accuracy of Buffy Coat in the Diagnosis of Disseminated Histoplasmosis in AIDS-Patients in an Endemic Area of Brazil"

_jof, 2019, doi:10.3390/jof5020047_

Round 1

Reviewer 1 Report

the authors present data on the use of Buffy coat smear and culture to diagnose disseminated histoplasmosis in patients with AIDS in an endemic area of Brazil. the diagnostic yield of the smear stain was relatively low, but culture added 11% to the sensitivity of the test. Buffy coat smears might be very useful in certain parts of the worlds, as they provides timely results where antigen testing is not available. the manuscript can be improved by the following:

- line 56: please provide a reference is available- otherwise state (unpublished data)

- line 88: were the culture results confirmed by additional testing (molecular probes)- if so please specify, if not please state that

- the Buffy coat culture was used as the gold standard against which the smear was compared. it is well known that blood cultures are negative in more than 25% of DH in AIDS. Therefore, it is reasonable to assume that up to 25 % of cases were missed based on culture. as a result the sensitivity of the smear was falsely elevated due to a smaller denominator (25% more could be diagnosed if other methods were used) - this fact needs to be acknowledged in the discussion as a limitation. 

- 3.3 Sequential testing: where patients were treated with antifungal agents between the repeated tests. if so how many were and did that alter the results of the repeated tests. if not were patients re-tested because they were clinically worse without antifungal therapy?

- Line 173: please provide a reference is available- otherwise state (unpublished data)

- Line 204-206: please describe whether patient received antifungal therapy between repeated tests. 

Author Response

Please, find attached the answers to the review. We addressed each question and comments, and highlighted in yellow the answer. Thank you for the suggestions and questioning , that were very pertinent and contributed to improve the article substantially.

Reviewer 2 Report

In low resources countries, having a simple and cheap method that enables the increase of the diagnosed cases of a lethal disease, it is a plus. Thus, I consider that this paper has its value and applicability and that it will be well received by clinicians and lab technicians of those countries.

However, some points need to be clarified or corrected:

Page 1, lines 37-40: The authors stated that “ histoplasmosis is considered the most common mycosis in humans”. That is not true! Did the authors mean that “ histoplasmosis is considered the most common ENDEMIC mycosis in humans”?

Page 1, introduction: the authors never refer to Histoplasma capsulatum var. duboisii. That should be mentioned since these yeasts are bigger and do not appear intracellularly.

Page 2, line 60: Please replace “…diagnostic tool for DH in HIV patients, the sensitivity and, specificity of this test in that specific…” by “…diagnostic tool for DH in HIV patients, and to determine the sensitivity and specificity of this test in that specific…”

Page 2, line 68: Please replace “…samples with culture results for fungus of HIV positive patients were included.” by “samples of HIV positive patients with cultural results for fungi were included”.

Page 2, line 76: Please replace “…fungus species led to exclude the samples from the analyses” by “…fungal species led to the exclusion of the samples from the present study”

Page 2, line 83: Please replace “…slide, that was stained by Giemsa for microscopic examination and the rest was inoculated for fungal…” by “…slide and stained by Giemsa for microscopic examination. The remaining was inoculated for fungal…”

How was the inoculation performed? Inoculated volume? With a loop?

Page 3, line 84: Sabouraud Agar or Sabouraud dextrose Agar?

Page 3, line 87-88: The authors stated that “cultures were considered as positive, if mycelium –phase as hyphae and tuberculate macroconidia were seen after staining with cotton blue…”

As the authors should know, Sepedonium sp. has a very similar micromorphology when compared to Histoplasma capsulatum. Did the authors confirm the Histoplasma identification by conversion to yeast phase to check dimorphism?

Page 3 lines 107-108: The authors never mentioned never refer how many buffy coats were positive without a positive culture

Page 3, line 111: Please replace “…18% for the Histoplasma positive cultures” by “18% for the subgroup of patients with Histoplasma positive cultures”

Page 4, line 132: Two of the 99 PATIENTS

Page 4, line 140: Please replace “Ten out of 12 growths in the first culture, continued positive in the second one…” by “Ten out of 12 cases had fungal growth in the first culture performed, remained positive in…”

Page 4, line 145: The authors stated that:” The buffy coat culture increased the prevalence of histoplasmosis in 11.1% of those with one test.” Compared to what? I believe that the authors wanted to say the following: “Performing more than one buffy coat’s culture increased the diagnosis of histoplasmosis in 11.1%”

Page 5, line 174: Please replace “rests” by “relies”

Page 5, line 220: The lower death RATE

Author Response

(The authors gave the same response as above.)

Reviewer 3 Report

In this manuscript, authors are establishing the utility of 'buffy coat' in the diagnosis of disseminated histoplasmosis in AIDS patients that may be of importance in poorly resourced areas. The work is of very important given the AIDS endemicity in many parts of the world where histoplasmosis is common. In general, the manuscript is well written and has included the pros and cons of the test. One major point that I would like authors to stress is to make it clear the cons of the test and strongly suggest the better alternatives in the abstract.

Minor: line #112. Should be "The CD4+ T lymphocyte count in the......."

Author Response

(The authors gave the same response as above.)

Round 2

Reviewer 2 Report

The authors aswered to all the issues raised and I do not have other points to refer.